# Negative Influence of Aging on Differentiation and Proliferation of CD8^+^ T-Cells in Dogs

**DOI:** 10.3390/vetsci10090541

**Published:** 2023-08-25

**Authors:** Akinori Yamauchi, Sho Yoshimoto, Ayano Kudo, Satoshi Takagi

**Affiliations:** 1Laboratory of Small Animal Surgery, Department of Veterinary Medicine, School of Veterinary Medicine, Azabu University, 1-17-71 Fuchinobe, Chuo-ku, Sagamihara 252-5201, Kanagawa, Japan; 2Department of Clinical Sciences and Advanced Medicine, School of Veterinary Medicine, University of Pennsylvania, Philadelphia, PA 19104, USA

**Keywords:** dog, aging, CD8^+^ T-cell, T-cell proliferation, T-cell subsets, T-cell senescence, immune senescence

## Abstract

**Simple Summary:**

Immunosenescence is known as changes in the immune system associated with aging and is said to be an important factor for cancer development and the therapeutic effect of cancer immunotherapy in humans. With the recent extension of the dog lifespan, cancer has become the leading cause of death in dogs; therefore, it is necessary to understand immune senescence in dogs. However, it is not well understood how aging impacts the differentiation status and cell proliferation of canine CD8^+^ T-cells, which play an important role in cancer immunity. In this study, we aimed to determine which subsets of canine CD8^+^ T-cells were influenced by aging and their proliferative potential. We stimulated CD8^+^ T-cells from older dogs using antibody-coated beads and interleukin-2 (IL-2) and found that the differentiation of central memory subsets into effectors was enhanced in older CD8^+^ T-cells, and the proliferative capacity of both effector and central memory T-cells was decreased. These results suggest that conventional stimulation with T-cell receptor ligands and IL-2 might not be sufficient for culturing T-cells from older dogs. Therefore, the selection of a less differentiated population or modifications of cytokine signaling might be needed to expand older dog CD8^+^ T-cells.

**Abstract:**

Immunosenescence is an age-related change in the immune system characterized by a reduction in naïve T-cells and an impaired proliferative capacity of CD8^+^ T-cells in older individuals. Recent research revealed the crucial impact of immunosenescence on the development and control of cancer, and aging is one of the causes that diminish the therapeutic efficacy of cancer immunotherapies targeting CD8^+^ T-cell activation. Despite dog cancer being defined as an age-related disease, there are few fundamental understandings regarding the relationship between aging and the canine immune system. Therefore, we aimed to elucidate the characteristics of immunosenescence in dogs and analyzed the effects of aging on the differentiation status and proliferation of canine CD8^+^ T cells using T-cell specific stimulation with anti-canine CD3/CD28 antibody-coated beads and interleukin-2. As a result, we found that older dogs have a lower proliferative capacity of CD8^+^ T-cells and a reduction in the naïve subset in their peripheral blood. Further analysis showed that older dogs had attenuated proliferation of the effector and central memory subsets. These results indicate the importance of maintaining less differentiated subsets to expand CD8^+^ T-cells in dogs and provide helpful insight into the development of dog immune therapies that require T-cell expansion ex vivo.

## 1. Introduction

Significant advancements in veterinary medicine have brought about remarkable progress in extending the lifespan of dogs, leading to an enhanced understanding of age-related diseases such as cancer, infections, and autoimmune disorders. Cancer has emerged as the leading cause of death in dogs, accounting for approximately one-fourth of all cases and affecting approximately 50% of dogs aged 10 years and older [1]. Therefore, there is an urgent need to elucidate the mechanisms underlying the increased incidence of cancer with aging and to develop innovative therapeutic approaches.

The immune surveillance mechanism, mainly by CD8^+^ T-cells, is considered crucial in the control of cancer occurrence and progression. Briefly, three main processes are involved in the elimination of cancer cells within the body. First, natural immune cells such as dendritic cells and macrophages destroy cancer cells and capture cancer antigens. Subsequently, these cells migrate to nearby lymph tissues and present the cancer antigens to the T-cells responsible for acquired immunity. The cancer antigen presentation leads to CD8^+^ T-cells being differentiated from naïve T-cells into effector T cells, acquiring direct and potent cytotoxic abilities against cancer cells [2]. Additionally, some CD8^+^ T-cells differentiate into memory subsets, which have a self-replicating capacity and contribute to the maintenance of the T-cell population through differentiation into effector T-cells [3]. In recent years, cancer immunotherapy has gained attention as a novel treatment modality, with many approaches targeting the activation of CD8^+^ T-cells [4]. Immune checkpoint inhibitors (ICIs) such as anti-PD-1/PD-L1 antibodies have demonstrated an overall response rate of 20–40% in patients with human melanoma [5,6]. For oral melanoma in dogs, anti-PD-L1 antibodies exhibited an objective response rate of 14.3% [7], and anti-PD-1 antibodies showed a rate of 19.0% [8]. Additionally, chimeric antigen receptor (CAR)-T cell therapy, which involves the ex vivo activation and genetic modification of T-cells derived from patients with cancer, has recently gained attention. CAR-T cell therapy has shown complete remission rates of approximately 80% in human acute lymphoblastic leukemia and other hematological malignancies [9], and six CAR-T cell therapies have been approved by the Food and Drug Association since 2017 [10]. In dogs, Mason et al. have made progress in the development of CAR-T cells for diffuse large B-cell lymphoma, successfully eliminating cancer cells in the patient’s blood [11]. However, recent researchers suggested that aging might be a negative factor for the therapeutic efficacy of cancer immunotherapy, including ICIs and CAR T-cell therapy [12,13,14,15]. One of the proposed causes is age-related changes in the immune system, called immunosenescence.

In humans and mice, immune senescence includes a decline in naïve and memory T-cells, impaired proliferation of CD8^+^ T-cells in response to antigen stimulation, and decreased T-cell receptor (TCR) repertoire diversity [16,17,18]. Relatively undifferentiated CD8^+^ T-cells, such as naïve and memory subsets, have high proliferative capacity and contribute to the expansion and maintenance of the T-cell population [18,19]. On the other hand, effector subsets have strong cytotoxic potential, but they lose cell proliferative capacity in the early phase, ultimately transitioning toward cellular death [20]. Thus, the decline of naïve CD8^+^ T-cells is considered one of the most important characteristics of immunosenescence [21,22] and may diminish the proliferative capacity of T-cells and CAR-T cells derived from older individuals [23,24].

In dogs, age-related changes in T-cell-mediated immunity have been demonstrated in the decreased proliferative capacity of peripheral blood mononuclear cells (PBMCs) upon stimulation with mitogens such as phytohemagglutinin (PHA), concanavalin A (Con A), and pokeweed mitogen (PWM) [25,26,27]. However, mitogens have been reported to non-specifically stimulate not only peripheral blood T-cells but also monocytes [28], and the direct impact of aging on the proliferative capacity of CD8^+^ T-cells remains to be elucidated. Moreover, there have been limited studies examining the differentiation status of CD8^+^ T-cells in older dogs, and it also remains unclear which differentiation subsets of older CD8^+^ T-cells expand or fail to expand in response to antigen stimulation.

Therefore, we stimulated CD8^+^ T cells isolated from older and young dogs with anti-canine CD3/CD28 antibody-coated beads and interleukin-2 (IL-2) and analyzed the relationship between their differentiation status and proliferative capacity within each subset.

## 2. Materials and Methods

### 2.1. Animals

All animal experiments were approved by the Animal Experiment Committee of Azabu University (Approval No. 230327-36). Healthy young and older Beagle dogs bred in the laboratory of a small animal surgery at Azabu University were included in this study. All dogs were defined as “healthy” based on biannual health examinations, including diagnostic imaging, pre-screening blood tests, and physical condition checks. Based on previous reports, dogs aged 1–3 years were defined as “young dogs” and those aged 8 years and above as “older dogs” [29]. Blood tests were performed using an automatic blood cell counter (Microsemi LC-660; Horiba, Kyoto, Japan). In this study, all young dogs were 2 years old, while the old group consisted of dogs with a median age of 12 years (range: 11–15 years). All dogs analyzed in this study were unneutered. For blood, CD4, and CD8^+^ T-cell subset analyses, we used four female dogs in the young group and one male and three female dogs in the older group. To analyze CD8^+^ T-cell subsets and cell proliferation, we utilized three female dogs in the young group and one male and two female dogs in the older group.

### 2.2. Cell Culture

Lymphoprep solution (STEMCELL Technologies, Vancouver, BC, Canada) was used to isolate peripheral blood from mononuclear cells. CD8^+^ T-cells were isolated using a modified procedure from a previously reported method [30] and it was confirmed that the separation efficiency was approximately 90% for all samples using an EC800 Flow Cytometry Analyzer (Sony Biotechnology, San Jose, CA, USA). Briefly, we used the EasySep APC Positive Selection Kit II (STEMCELL Technologies) and an allophycocyanin (APC)-conjugated anti-canine CD8a antibody (YCATE55.9; Thermo Fisher Scientific, Waltham, MA, USA). CD8^+^ T-cells were cultured in a RPMI-1640 medium (Fujifilm Wako, Osaka, Japan) and supplemented with 100 unit/mL penicillin, 100 µg/mL streptomycin, 0.25 µg/mL amphotericin B (Antibiotic-Antimycotic; Thermo Fisher Scientific), 25 mM HEPES (Thermo Fisher Scientific), 55 mM 2-ME (Fujifilm Wako), and 10% fetal bovine saline (FBS) (Sigma Aldrich, Saint Louis, MO, USA) for up to 16 days. Then 0.5–1.0 × 10⁶ CD8^+^ T-cells were placed on a 48-well multi-dish for floating cells (Thermo Fisher Scientific) and activated using a modified procedure from a previously reported method [30]. Briefly, canine CD8^+^ T-cells were stimulated using Dynabeads M-280 Tosylactivated (DB14204; Thermo Fisher Scientific) coated with anti-dog CD3 (CA17.2A12; Bio- Bio-Rad Laboratories, Hercules, CA, USA) and anti-dog CD28 antibodies (1C6, Absolute Antibody, Upper Heyford, Somerset, UK) for 8 days at a 1:3 cell: bead ratio. Following the stimulation, from day 2 of culturing, RPMI-1640 containing 20 ng/mL recombinant human IL-2 (PeproTech, Cranbury, NJ, USA) was changed every other day (Figure 1).

### 2.3. Flow Cytometry

Individual antibodies were purchased from each company: fluorescein isothiocyanate (FITC)-conjugated antibodies for dog CD3 (CA17.2A12; Bio-Rad Laboratories), R-phycoerythrin-conjugated antibodies for dog CD4 (YKIX302.9; Bio-Rad), APC-conjugated antibodies for dog CD8 (YCATE55.9; Thermo Fisher Scientific), Alexafluour700-conjugated antibodies for dog CD3 (CA17.2A12; Novus Biologicals, Englewood, CO, USA), FITC-conjugated antibodies for dog CD44 (YKIX337.8; Bio-Rad Laboratories), and PE-conjugated antibodies for human CD62L (FCM46; Bio-Rad Laboratories). Then 0.5–1.0 × 10⁶ cells were collected, stained at 4 °C for 45 min, and fixed with 4% paraformaldehyde at 4 °C for 1 h. All samples were washed three times with Dulbecco’s phosphate buffered saline (Fujifilm Wako, Osaka, Japan), containing 1% FBS and 0.1% sodium azide and acquired on an EC800 Flow Cytometry Analyzer (Sony Biotechnology, San Jose, CA, USA). Viable cells were gated using forward- and side-scattered light.

### 2.4. Statistical Analysis

General blood test results and pre- and post-stimulation T-cell differentiation statuses of young and older dogs were compared using an unpaired *t*-test. The proliferation of CD8^+^ T-cells was analyzed using a one-way analysis of variance. All tests were performed using GraphPad Prism 9 software (GraphPad Software, Boston, MA, USA). Results were considered significant when the *p*-value was less than 0.05.

## 3. Results

### 3.1. CD8^+^ T-Cells from Older Dogs Had Lower Proliferative Capacity than Young Dogs

First, we analyzed the T-cell subsets in the peripheral blood, which showed no differences in CD4 and CD8 expression in CD3^+^ cells (Appendix A) between young and older dogs.

To clarify the effect of aging on cell proliferative capacity, we isolated CD8^+^ T-cells from the peripheral blood of each group and stimulated them with anti-canine CD3/CD28 antibody-conjugated beads and IL-2 (Figure 1). The addition of the beads and IL-2 enhanced CD8^+^ T-cell proliferation in both young and older dogs from day 0 to day 8 after culturing (fold change in cell number vs. day 0; young: 4.28 ± 0.21, older: 2.80 ± 0.15; Figure 2). However, older dogs had significantly lower proliferative capacity than young dogs from day 8 to day 16 after the stimulation (fold change in cell number vs. day 0; young: 17.50 ± 0.85, older: 6.66 ± 1.10; Figure 2).

### 3.2. In Older Dogs, the Population of Naïve CD8^+^ T-Cells Was Reduced, and the Expansion of Effector and Memory Subsets Was Attenuated

To elucidate the relationship between the differentiation status of CD8^+^ T-cells and the proliferation impairment observed in older dogs, we analyzed CD44 and CD62L expression using flow cytometry. We found that older dogs had fewer CD44^−^/CD62L^+^ (naïve) subsets (young: 57.28 ± 6.29%, older: 22.73 ± 1.74%) but more CD44^+^/CD62L^−^ (effectors) subsets (young: 14.20 ± 3.94%, older: 45.25 ± 3.61%, Figure 3a,b). We also analyzed the differentiation status of days 8 and 16 post-stimulation. We found that the differentiation status of both groups was the same on day 8 (Appendix A). However, on day 16, older dogs tended to have a lower percentage of CD44^+^/CD62L^+^ (central memory) subsets (young: 22.41 ± 6.69%, older: 7.96 ± 0.77%), and a tendency for more CD62L^−^/CD44^+^ (effectors) subsets was observed (young: 47.60 ± 7.71%, older: 67.42 ± 1.74%, Figure 3a,c).

To assess changes in the CD8^+^ T-cell subsets after stimulation, the difference between pre- and post-stimulation percentages was analyzed on days 8 and 16 of culturing. The data for the CD8^+^ T-cells of young dogs showed a greater reduction in the proportion of CD44^−^/CD62L^+^ (naïve) subsets (young: −33.35 ± 4.49%, older: −8.74 ± 3.99%, Figure 4) but less reduction in the proportion of CD44^+^/CD62L^+^ (central memory) subsets on day 16 (young: −1.36 ± 4.16%, older: −20.61 ± 3.20%, Figure 4). We did not observe any significant differences on day 8 (Appendix A). The CD44^+^/CD62L^−^ (effectors) subset was increased in both groups, but no significant change was observed on day 16 (young: 33.40 ± 4.42%, older: 22.17 ± 2.07%, Figure 4). 

We also evaluated the fold change in cell number for each differentiation state. As a result, older dogs had a lower proliferative capacity in the CD44^−^/CD62L^+^ (naïve) subset on day 8 (fold change in cell number; young: 3.28 ± 0.44, older: 1.51 ± 0.20, Appendix A) as well as in the CD44^+^/CD62L^+^ (central memory) (fold change in cell number; young: 15.84 ± 3.66, older: 1.88 ± 0.28, Figure 5) and CD44^+^/CD62L^−^ (effectors) subsets compared with young dogs on day 16 (fold change in cell number; young: 67.01 ± 8.48, older: 10.22 ± 2.08, Figure 5).

## 4. Discussion

The aim of this study was to elucidate the relationship between changes in the differentiation subsets of CD8^+^ T-cells in older dogs and their ex vivo cell proliferation. Initially, CD8^+^ T-cells were isolated from young and older dogs and expanded and cultured using antibody-coated beads and IL-2. An analysis of the CD8^+^ T-cell subsets revealed that the naïve CD8^+^ T-cell population in the peripheral blood of older dogs decreased compared with young dogs, and the effector and central memory subsets became the predominant CD8^+^ T-cell populations. Furthermore, after stimulation, the proliferation of CD8^+^ T-cells was attenuated in older dogs. Therefore, we compared the post-stimulation CD8^+^ T-cell subsets between young and older dogs. As a result, in young dogs, the proportion of the naïve subsets decreased after stimulation, but there was less change in the proportion of the central memory subsets compared with older dogs. In contrast, in the older group, there was a significant decrease in the proportion of central memory subsets after stimulation, indicating the induction of differentiation towards the effectors. Further analysis of the cell proliferation for each subset revealed that the proliferation of both the effector and central memory subsets was significantly reduced in older dogs compared with young dogs (Figure 6).

In this study, it became evident that achieving cell proliferation in CD8^+^ T-cells from the peripheral blood of older dogs, containing effector and memory subsets, was challenging and comparable to those derived from young dogs. In humans and mice, the reduction in naïve T-cells in peripheral blood and the decline in the proliferative capacity of CD8^+^ T-cells are common characteristics of immunosenescence [22,31,32,33]. In this study, we observed a decrease in naïve CD8^+^ T-cells in the peripheral blood of older dogs, consistent with previously reported results in dogs [34]. Typically, naïve T-cells stimulated by antigens differentiate into memory subtypes and eventually lose self-renewal ability through repeated cell divisions and further differentiation into effector subtypes [20,35]. We observed little difference in the differentiation status of CD8^+^ T-cells between young and older dogs after stimulation but found differences in the rate of change and the relative increase in cell numbers in each subset. In young dogs, the proportion of the naïve subsets decreased after stimulation, and the proportion of central memory subsets was maintained. On the other hand, in older dogs, there was a significant decrease in the proportion of central memory CD8^+^ T-cells, and the cell proliferation in both the effector and central memory subsets was attenuated. These findings suggest that the proportion of naïve CD8^+^ T-cells before stimulation may be critical for maintaining the memory subsets after ex vivo stimulation in dogs. Therefore, the decrease in the naïve subset in older dogs may have limited the expansion of the T-cell population, including effector and central memory CD8^+^ T-cells.

In this study, we first isolated CD8^+^ T-cells from PBMCs and then used anti-canine CD3/CD28 antibody-coated beads as T-cell-specific activators. There were two main objectives for employing this method: first, to evaluate directly the differentiation and cell proliferation of CD8^+^ T-cells in older dogs, and second, to contribute to the development of T-cell infusion therapy targeting older cancer-bearing dogs. Previous studies used Con A to stimulate PBMCs for evaluating T-cell proliferation [25,34]. However, Con A binds to cell-surface carbohydrates, glycoproteins, and glycolipids, inducing non-specific immune stimulation regardless of species [36]. Additionally, Con A induces various immune cellular activities, not only in T-cells but also in macrophages and B-cells [36,37]. Furthermore, while Con A is useful for assessing the short-term cell phenotype after addition due to its robust T-cell proliferation activity, its transient effect makes it less suitable for stimulating the large quantities of T-cells required for cell infusion therapies [38]. Moreover, Con A has been shown to cause kidney and autoimmune liver disorders in mice, raising safety concerns when using it as a stimulation agent in cell-based therapies intended to be reintroduced into the body [39,40,41]. The magnetic-beads cell separation method used in this study was modified based on a previous report [30], and was found to achieve approximately a 90% efficiency in isolating CD8^+^ T-cells from canine PBMCs during the preliminary investigations (data not shown). Additionally, the anti-CD3/CD28 antibody-coated magnetic beads induce TCR-dependent cell signaling through CD3/TCR complexes and co-stimulatory signaling through CD28, leading to specific and long-term T-cell proliferation [42]. Moreover, the stimulating beads can be removed from the T-cell suspension using a magnet, making them clinically useful [43]. Recently, this bead-based T-cell stimulation protocol has been applied in human CAR-T cell therapy [44,45]. Mason et al. also adopted this protocol for T-cell stimulation during dog CAR-T cell production [11,46]. Furthermore, additional research has shown the usefulness of the beads and IL-2 stimulation protocol for the proliferation and culturing of canine peripheral blood T-cells [42]. Thus, in this study, we believe that this method made it simpler to evaluate the changes in CD8^+^ T-cell differentiation subsets and cell proliferation in older dogs.

One of the limitations of this study is the lack of a comprehensive understanding of the specific mechanisms by which aging decreases the proliferation capacity of canine T-cells. It has been reported that the specific protein expressions and downstream signaling pathways involved in T-cell proliferation decline with age. For instance, the expression of CD25, the IL-2 receptor subset on CD8^+^ T-cells, is crucial for IL-2-induced cell proliferation and may be necessary for maintaining immune responsiveness in older individuals [47]. Previous studies in humans have suggested that a decreased expression of CD25, along with CD28 as a co-stimulatory molecule, in older individuals contributes to reduced cell proliferation capacity [48]. Interestingly, our results showed that beads alone did not induce the proliferation of CD8^+^ T-cells in young or older dogs. This suggests that the proliferation of canine CD8^+^ T-cells strongly depends on the presence of IL-2. Moreover, the differences in T-cell proliferation between young and older dogs became evident upon IL-2 supplementation, suggesting a potential weakening of the signaling related to cell proliferation in older dogs at the time of IL-2 addition. For future studies, analyzing CD25 expression in older dog CD8^+^ T-cells and evaluating changes in intracellular signaling activity, such as in the JAK/STAT pathway, through IL-2 titration is needed to elucidate the mechanism of decreased cell proliferation in older dog CD8^+^ T-cells. Therefore, alternative cytokine signals to IL-2 and epigenetic regulation to support cell proliferation might be helpful for the cultivation of CD8^+^ T-cells derived from older dogs.

One of the aims of this study was to contribute to the development of cancer immunotherapy, particularly CAR-T cell therapy, targeting older dogs with cancer. However, the CD8^+^ T-cells used in this study from dogs were not transduced with CAR vectors, and further investigations are needed to understand the impact of aging on CAR-T cell therapy in dogs. Studies in humans and mice have shown that aging may limit the therapeutic efficacy of CAR-T cell therapy. In human studies, CAR-T cells derived from older individuals were found to have reduced proliferative capacity [24]. Moreover, CAR-T cells generated from older mice showed superior cytotoxicity but had shorter persistence and were less likely to exhibit memory phenotypes [49]. Considering that dogs may mimic the immunosenescence observed in humans and mice, it is reasonable to speculate that the therapeutic effects of CAR-T cells may diminish with age in dogs as well. Recent reports on humans have demonstrated that CAR-T cells generated from relatively undifferentiated T-cells, such as the naïve or memory phenotypes, exhibit higher antitumor efficacy compared with bulk-derived CAR-T cells [19]. Therefore, in the development of CAR-T cell therapy targeting older dogs with cancer, it might be essential to assess the differentiation state of T-cells before CAR vector transduction, and enriching less differentiated T-cells seems to improve therapeutic efficacy.

Lastly, canine cancer has been considered a valuable model for spontaneous cancer development due to its similarities to humans in terms of histology, biological behavior, and etiology, supporting the development of cancer therapies in human medicine [50]. In line with previous research, our results suggest that the immune system in older dogs may partially mimic immunosenescence observed in humans. Hence, further research may support the development of translational cancer immunotherapies not only for older dogs but also for older persons.

## 5. Conclusions

Our results showed that CD8^+^ T-cells derived from older dogs were unable to efficiently induce cell proliferation compared with those derived from young dogs when stimulated ex vivo. In particular, the proliferation of the effector and central memory subsets was diminished in older dogs. Our results suggest that the differentiation status of CD8^+^ T-cells or sensitivity to IL-2-dependent stimulation plays an important role in CD8^+^ T-cell expansion. Therefore, the cultivation of CD8^+^ T-cells derived from older dogs implies the need to select a less differentiated population or modify the cytokine signaling involved in cell proliferation.

## Figures and Tables

**Figure 1 vetsci-10-00541-f001:**
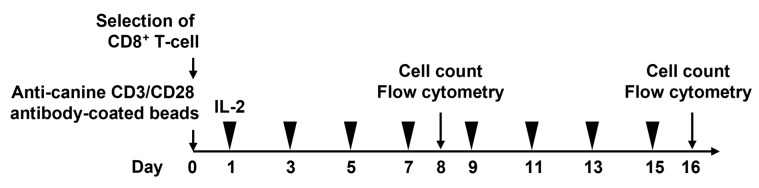
Schedule for CD8^+^ T-cell stimulation. CD8^+^ T-cells are isolated from young and older dogs and cultured with anti-canine CD3/CD28 antibody-coated beads daily. Then, interleukin-2 (20 ng/mL) is added to the medium. The medium is changed every other day (black arrowheads). All cells are counted and analyzed on days 8 and 16.

**Figure 2 vetsci-10-00541-f002:**
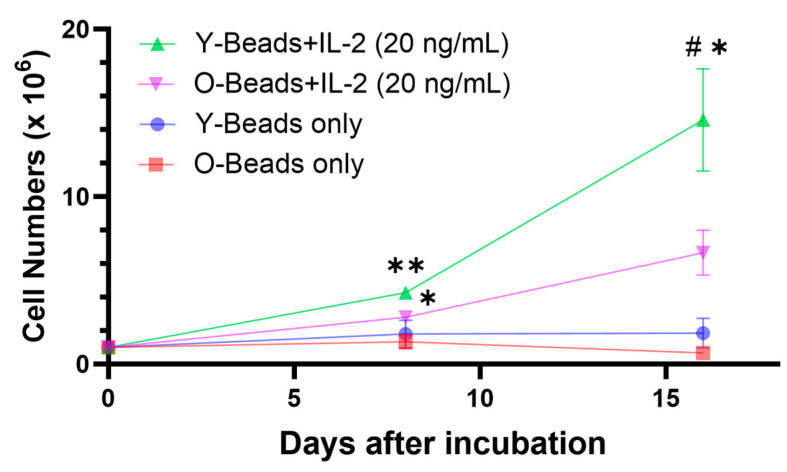
Cell proliferation of young and older CD8^+^ T-cells after culturing with anti-canine CD3/CD28 antibody-coated beads and IL-2. Y: young dogs; O: older dogs. * *p* < 0.05, ** *p* < 0.01 compared with day 0 of each group. # *p* < 0.05 compared with day 16 of young dogs: beads + IL-2. n = 3 for each group. IL-2: interleukin-2.

**Figure 3 vetsci-10-00541-f003:**
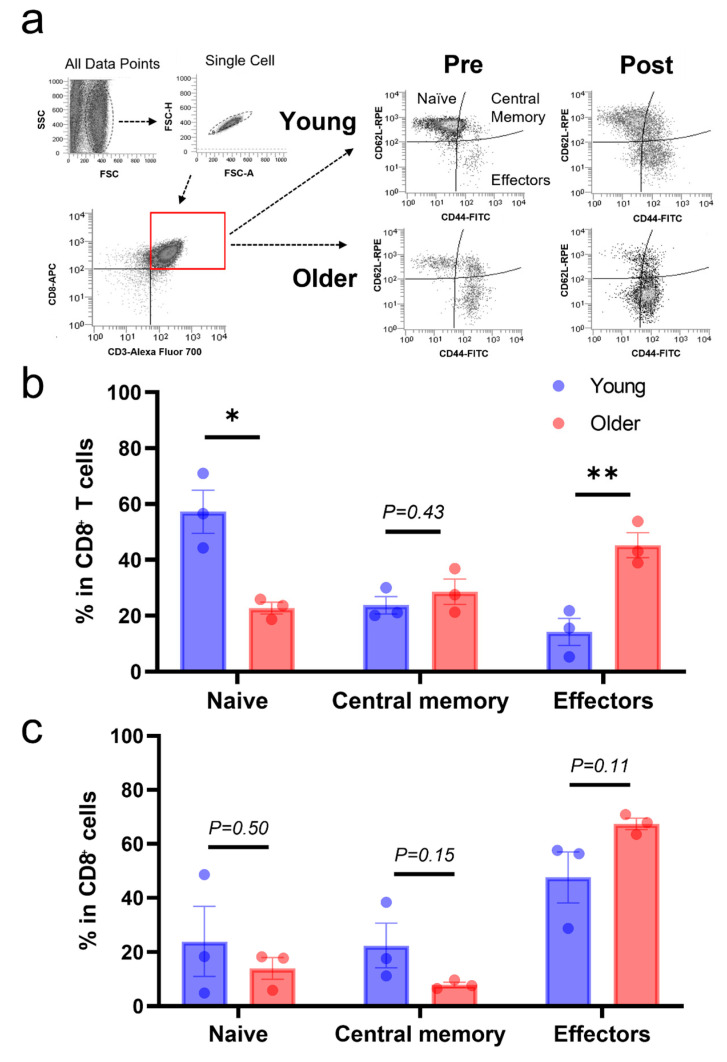
(**a**) Representative flow cytometric analysis of CD44 and CD62L expression in CD8^+^ T-cells pre- and post-stimulation. (**b**) Percentage of CD8^+^ T-cell subsets pre-stimulation. (**c**) Percentage of CD8^+^ T-cell subsets on day 16 post-stimulation. Each plot represents individual data. * *p* < 0.05, ** *p* < 0.01. n = 3 for each group.

**Figure 4 vetsci-10-00541-f004:**
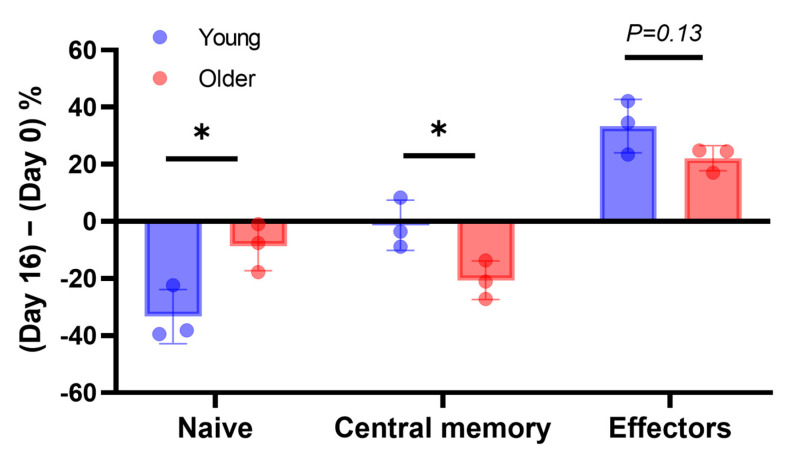
Percentage changes in subsets in CD8^+^ T-cells from day 0 to day 16 post-stimulation. Each plot represents individual data. * *p* < 0.05. n = 3 for each group.

**Figure 5 vetsci-10-00541-f005:**
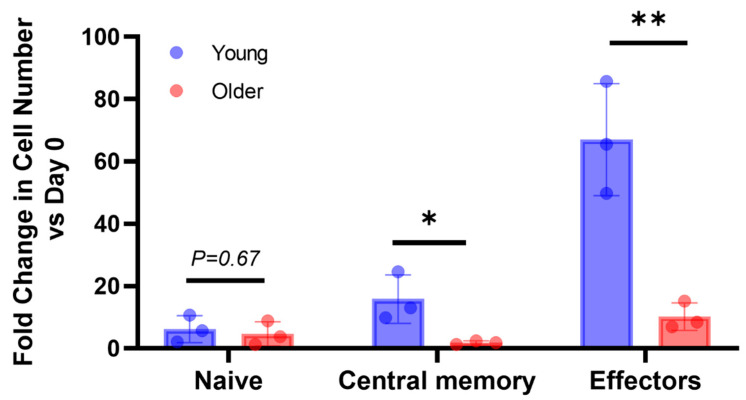
Fold changes in cell numbers in each CD8^+^ T-cell subset on day 16 post-stimulation compared with pre-stimulation. Each plot represents individual data. * *p* < 0.05, ** *p* < 0.01. n = 3 for each group.

**Figure 6 vetsci-10-00541-f006:**
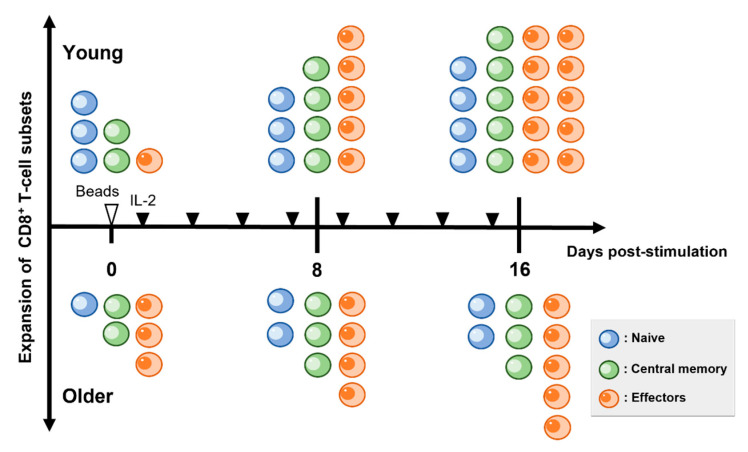
Schematic diagram of the results of this study. Cell proliferation of effector and central memory CD8^+^ T-cells was attenuated in older dogs after culturing with anti-canine CD3/CD28 antibody-coated beads and IL-2.

## Data Availability

All data supporting the findings of this study are available from the corresponding author upon request.

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
