# Peer review of "Negative Influence of Aging on Differentiation and Proliferation of CD8+ T-Cells in Dogs"

_vetsci, 2023, doi:10.3390/vetsci10090541_

Round 1
Reviewer 1 Report
This study utilizes young and old purpose-bred dogs to study the differences in ex vivo expansion of CD8+ T cell subsets occurring with age. The topic is relevant and warrants investigation, however the authors overemphasize the impact of their work on CART cell development without a clear explanation of how their findings can be used to optimize CART development. There are also a few key methodological features that require further explanation/validation.
Introduction:
Line 58-60: These references do not support this statement. Since this statement is repeated multiple times throughout the article I think its important that this is referenced appropriately.
Line 71-72: Ref 7 does not identify age as a negative prognostic factor and ref 15 is a mouse model study, so these references do not support the claim, “However, the number of naïve T-cells in the peripheral blood decreases in older 70 patients [13,14], which may lead to a reduction in the therapeutic response and persistence of CART-cell therapy in older patients with cancer [7,15].”
Line 74-75 – This sentence needs to be rearranged to be correct. I think the authors perhaps intended to say something similar to the following? “A few studies on canine T-cell senescence have shown that the proliferative capacity of T-cells exposed to Concanavalin A is reduced in older dogs. Concanavalin A triggers nonspecific stimulation of peripheral blood T lymphocytes and monocytes.”?
Methods:
Can you provide confirmation of ethics committee approval?
Please provide the sex, intact/neutered status of the animals included.
It is not clear how many dogs were included. Suppl. Fig. 1 seems to show 4 dogs in each group, but figures from the rest of the study only show “n = 3”. This should be made clear in the methods how many dogs were included in each assay.
Line 103: What clone of CD8 was used for this?
Was the efficacy of the bead separation confirmed by flow cytometry either at baseline or either of the 2 subsequent timepoints? I.e. how do you know your methods led to an enriched/pure population of CD8+ T cells? Have these methods been published previously and showed efficacy of the bead separation?
Line 112: how many hours? Do you mean a full 24 hours or just during one working day (~8hrs)?
Line 128: which lasers is the machine configured with?
Results:
Flow cytometry was performed on day 8 as well per methods? Please show these data either in this figure or as supplemental data.
Line 186-191: Reference these statements.
Line 195-196: Specifically, “…canine CD8+ T cell subsets…” not “…canine T cell subsets…”.
Figure 2: I don’t understand what each of the asterisks and the hash mean on this figure. Please clarify.
Love figure 6!
Discussion:
The authors argue that understanding ex vivo lymphocyte expansion is necessary for adopting CART therapy in older dogs. They also state that previous studies have shown reduced expansion of T cells in older dogs but it remains unclear which subsets have impaired expansion. They should also explain how understanding which specific subsets are expanding vs. not in older dogs will help to optimize T cell expansion for CART therapy. This link is presently missing from their argument.
The authors should also discuss how this information gathered in this study can now be used to optimize CART development in dogs…would they propose sorting a particular cell type instead of bulk CD8 T cells for CART development, and/or would they expect decreased efficacy/increased toxicity in older dogs compared to younger dogs given the preferential expansion of effectors? Please elaborate.
Reviewer 2 Report
The authors present a study focusing solely on the in vitro expansion of CD8+ T lymphocytes in dogs. However, it is important to note that the lymphocytes in the study were not transduced with any CAR vector. Therefore, references to CAR therapy or CAR T lymphocytes should be eliminated from the abstract and introduction sections, reserving them for discussion purposes only.
Furthermore, there are several inaccuracies in the references cited throughout the manuscript. For instance, references 16, 17, 18, and 31 appear to be incorrect. I recommend a thorough review and verification of all references to ensure their accuracy and reliability.
Additionally, the article aims to investigate differences between lymphocytes from young and old dogs. However, the age range of the donors is not mentioned, nor is there clarification regarding what age range constitutes "young" or "old" in this context. It is essential to provide this information.
Moreover, the title of section 3.2 does not appear to accurately reflect the findings obtained. I suggest revisiting the title to align it more closely with the results presented.
Lastly, the fold change values in Figure 5 do not align with the results shown in Figure 2. This discrepancy raises concerns about the accuracy of the data presented.
In conclusion, I believe that further detailed studies are required to draw valid conclusions from the presented results.
Reviewer 3 Report
This study by Yamauchi et al investigates the proliferative potential of CD8 T cells isolated from young or old dogs. The authors identify a diminished proliferative potential for CD8 T cells from older dogs upon stimulation with CD3/CD28 beads and IL2 supplement. The authors conclude that further investigations in the underlying mechanisms leading to a diminished proliferative potential will be important for the development of CAR T cell therapies for older dogs, more likely to be affected by cancer.
The study is well documented, and the results are clearly reported. One puzzling aspect is the focus solely on CD8 T cells, which is not explained in the manuscript, other than a brief mention that analysis of PBMC proliferation would not allow a precise quantification of T cells proliferation. CD4 T cells are also very important for CAR T cells therapy, therefore the proliferative potential of CD4 T cells would be equally interesting.
This work is focused on the differential analysis of proliferation between older and younger dogs. It would have been interesting to see if a titration of IL2 could boost proliferation of T cells in older dogs, but obviously it was out of the scope of this investigation.
In conclusion, the data presented is very limited, but might be nonetheless of interest to the readership of Veterinary Science. It would greatly benefit, if the focus would be expanded to CD4 T cells as well.
Minor comments:
L55 80% ... OS is more like 60% …
L64 “Additionally, it is crucial to maintain the number of cultured T-cells in vivo after their administration to patients” not clear what it should mean.
L76 Ref16 is about non specific stimulation of macrophages. From mice, not from dogs. What are the references for the studies in older dogs?
L80 Ref 16 has been probably mistakenly added here.
L82 Ref 18 seems to be wrong: sentence3 starts with Mason et al. reference is from Wang et al.
L103 please indicate clone number, cat. No.
L105 antibiotic-antimycotic ? please specify
L109 again Ref 18 is the wrong one.
L109 concentration of 3/28 coated beads? Where beads removed?
L140 In suppl Fig 1 CD4 and CD8 data are shown but not CD3. It is not clear to me if the results are from triple stained T cells gated for CD3 or else. The total of CD4 and CD8 seems to be less than 100%. Are there CD4 and CD8 negative T cells? Percentage? The raw data would help with the interpretation.
L215 Ref 30 should go to line 212 with Refs 27-29, Ref 31 should be cited here
L222 Ref 18 is about human and not canine T cells (ref19?) also refs 31-33 are shifted by 1 (32-34) etc.
Supplementary Fig. 2
Round 2
Reviewer 2 Report
The article has been improved according to the suggestions made
Reviewer 3 Report
The authors have addressed all the major issues and discussed the limitations of this study. I hope the authors will continue their effort to characterize more in detail this important aspects.